# Oral characteristics and dietary habits of preterm children: A retrospective study using National Health Screening Program for Infants and Children

**Lan Herr[1]☯, Juhyun Chung[2]☯, Ko Eun Lee[3]‡, Jung Ho Han[4]‡, Jeong Eun Shin[4]‡, Hoi-In Jung[2]\*, Chung-Min Kang[1]\***

1 Department of Pediatric Dentistry, Yonsei University College of Dentistry, Seoul, Republic of Korea,
2 Department of Preventive Dentistry & Public Oral Health, Yonsei University College of Dentistry, Seoul, Republic of Korea, 3 Department of Pediatric Dentistry, Kyung Hee University Dental Hospital, Seoul, Republic of Korea, 4 Department of Pediatrics, Yonsei University College of Medicine, Seoul, Republic of Korea

☯ These authors contributed equally to this work.
‡ KEL, JHH and JES also contributed equally to this work.
\* kangcm@yuhs.ac (CMK); junghoiin@yuhs.ac (HIJ)

**Data Availability Statement:** Data cannot be shared with the public owing to restrictions. The "infant medical screening cohort database" can be accessed with Korean NHIS's (National Health

## Abstract

The rate of preterm birth is increasing worldwide and preterm infants are susceptible to oral health problems. Hence, this study aimed to investigate the effect of premature birth on dietary and oral characteristics as well as dental treatment experiences of preterm infants using a nationwide cohort study. Data was retrospectively analyzed from National Health Screening Program for Infants and Children (NHSIC) of the National Health Insurance Service of Korea. 5% sample of children born between 2008 and 2012 who completed first or second infant health screening were included and divided into full-term and preterm-birth groups. Clinical data variables such as dietary habits, oral characteristics, and dental treatment experiences were investigated and comparatively analyzed. Preterm infants showed significantly lower rates of breastfeeding at 4–6 months ($p<0.001$), delayed start of weaning food at 9–12 months ($p<0.001$), higher rates of bottle feeding at 18–24 months ($p<0.001$), poor appetite at 30–36 months ($p<0.001$) and higher rates of improper swallowing and chewing function at 42–53 months ($p = 0.023$) than full-term infants. Preterm infants also had eating habits leading to poor oral conditions and higher percentage of absence of dental visit compared to full-term infants ($p = 0.036$). However, dental treatments including 1-visit pulpectomy ($p = 0.007$) and 2-visit pulpectomy ($p = 0.042$) significantly decreased when oral health screening was completed at least once. The NHSIC can be an effective policy for oral health management in preterm infants.

## Introduction

The World Health Organization (WHO) defines preterm birth as birth before 37 weeks of pregnancy. Every year, approximately 15 million babies are born preterm, and this number is

Insurance Service, https://nhiss.nhis.or.kr) approval. Researchers who meet the accessibility criteria can use this confidential data. The authors had no privileges related to these criteria compared to others.

**Funding:** This work was supported by the National Research Foundation of Korea(NRF) grant funded by the Korea government (MSIT) (No. 2020R1G1A1100275) and the Bio & Medical Technology Development Program of the National Research Foundation & funded by the Korean government (MSIT) (No. 2022M3A9F3016364). The funders had no role in study design, data collection and analysis, decision to publish, or preparation of the manuscript.

**Competing interests:** The authors have declared that no competing interests exist.

increasing worldwide [1]. According to WHO, across 184 countries, the rate of preterm birth ranges from 5% to 18% of babies born. The birth of preterm newborns represents an enormous global problem with increased economic, social, family, and individual costs. Preventive and health-promoting measures are needed to improve the quality of life of these infants.

Preterm babies often have immature neural, cognitive, cardiovascular and gastrointestinal systems and usually develop long-term complications such as cerebral palsy, impaired vision and hearing, behavioral and psychological problems, and chronic health issues [2]. A preterm baby is likely to be admitted to the neonatal intensive care unit (NICU) for a long period. The American Academy of Pediatrics lists independent oral feeding as one of the criteria for discharge [3]. A reason for late hospital discharge is often the inability of the baby to engage in safe and efficient oral feeding [4]. The development of oral feeding skills requires coordination among the infant's abilities to suckle, swallow, and breathe, and these skills are usually learned during breast and bottle feeding. Safe oral feeding entails proper oxygenation; however, preterm babies have short respiratory rates that do not allow sufficient time for breathing between swallows, and this may result in apnea [5]. The readiness of a preterm infant to start oral feeding is assessed using the Preterm Oral Feeding Readiness Scale (POFRAS), which has an accuracy of 71.29% [6], but its reliability still needs to be assessed [7]. Recently, non-nutritive sucking and swallowing exercises have been introduced in routine NICU care to hasten the start of oral feeding and ensure smooth weaning from tube feeding to oral feeding [8].

In the early neonatal period, preterm babies are often cared for by the medical team in the NICU and not by the mother or father as in the case of the majority of full-term babies. Prolonged tube feeding in the NICU and lagged transition to oral feeding are associated with not only high medical costs and aversion to oral feeding but also poor attachment between the infant and its parents. The incidence of postnatal depression and stress levels were significantly higher among parents of preterm babies admitted to the NICU than among parents of full-term babies [9]. When an infant is in the NICU for a prolonged period, the parents feel confused, struggle with negative feelings about the situation, and lose the motivation to play the role of parents [10]. Hence, early interventions from healthcare teams and social workers are necessary not only to promote the physical and emotional attachment of parents to their infants but also to prepare parents for transition into parenthood and the task of taking care of the baby after discharge.

Similar to other parts of the body, the oral structures are affected by premature birth. The most prevalent oral complications are hypoplasia and opacities of the dental enamel in primary teeth [11–14]. Several studies have reported the deleterious effects of preterm birth on oral health and development, such as crown dilaceration [15], palatal distortions [16], delayed tooth eruption [17], and dental caries [18]. Immature neurodevelopment and endotracheal intubation may cause dental occlusion and influence jaw symmetry [16, 19]. Early nutritional support is important for preterm infants because it influences long-term health and development [20], can reduce the risk of impaired growth, and can limit the need for high levels of nutrient supplementation after discharge. However, premature infants may be fed more frequently to compensate for the growth delay. The infants may also be prone to feeding at night or retaining food in the mouth for a long time owing to deficiency in the suck-swallow-breathe action [21]. These poor dietary habits can adversely affect the long-term oral health and hygiene of premature babies.

Very few high-quality longitudinal studies have focused on oral characteristics of children born preterm. Furthermore, few studies have tracked the changes in dietary habits and oral health over time. To our knowledge, this is the first nationwide retrospective cohort study on the oral health of children born preterm. The main aim of this study was to determine the effect of premature birth on the dietary habits and oral structures and the rate of utilization of dental care services in this population.

## Methods

### Ethical considerations

The study was reviewed and approved by the institutional review board (IRB No 2-2019-0045), and the requirement for informed consent was waived because we used de-identified administrative data (NHIS-2021-2-104) made available by the Korean National Health Insurance Service (NHIS).

### Study data

In South Korea, newborn babies undergo National Health Screening Program for Infants and Children (NHSIC) covered under national health insurance [22]. Seven health screening sessions are provided between the ages of 4 and 71 months: at 4–6 (first), 9–12 (second), 18–24 (third), 30–36 (fourth), 42–48 (fifth), 54–60 (sixth), and 66–71 (seventh) months. Three oral screening sessions are provided: at 18–29 (first), 42–53 (second), and 54–65 (third) months. The NHIS collects the screening data in its infant medical and oral screening cohort database [23]. The available data provided by NHIS is 5% of infants born between 2008 and 2012 who underwent at least one of the first and second health screening examinations (n = 84,005). Data is analyzed by remote access to a virtual computer only for a certain period of permitted time. The infants who complete screening are coded by personal identification number and saved in the database. There are 239 items in health screening and 98 items in oral screening database. Data on social and economic variables such as age, location, type of insurance, income level, and disability were collected; data on clinical status such as type of facility/establishment, equipment, and locations were also collected.

### Selected study variables

Data of clinical variables such as dietary habits, oral characteristics, and dental experiences, as specified in NHSIC records, were analyzed. The medical and oral screening consisted of questionnaire survey answered by the infants' parents (Table 1) as well as by the examining physician and dentist. The questionnaire assessed the health and oral health-related characteristics and dietary habits of infants. The clinical and dental experiences of children born preterm and full-term were compared using the frequency of the use of different dental treatments, such as the number of dental clinic visits, radiographic examination, tooth extraction, pulp treatment, glass ionomer restoration, and sealant application.

### Study participants

This study included 5% of children born between 2008 and 2012 who completed the first or second infant health screening provided by the NHIS (n = 84,005) (Fig 1). Participants were grouped under full-term or preterm categories, as indicated by the questionnaire response of the parents. These groups were sub-divided according to the birth weight specified in the questionnaire. Low-birth-weight participants were divided into three sub-categories: extremely low birth weight (ELBW, birth weight less than 1000 g), very low birth weight (VLBW, birth weight between 1000 g and 1500 g), and low birth weight (LBW, birth weight between 1500 g and 2500 g). Premature birth was answered by yes or no survey question in screenings however, the birth week (gestational age) was not recorded. Thus, birth weight distribution was utilized to evaluate the analyses. Infants whose birth weight was missing, was below 500 g, or was above 5000 g were excluded (n = 17,632). For dental treatment analysis, the remaining 66,373 (79%) infants were included. Among them, 27,200 infants who had not finished fifth infant health screening were excluded for multivariable logistic regression analysis on association of

**Table 1. Health and oral health screening questionnaire responses.**

| Screening type | Session | Age | Screening questionnaire |
|---|---|---|---|
| Health screening | 1st | 4–6 months | Has your newborn been admitted to NICU for more than 5 days after birth? |
| | | | What was the birth weight of your newborn? |
| | | | Do you breastfeed or formula feed? |
| | 2nd | 9–12 months | When did your child start first solid foods (weaning)? |
| | | | Does your child sleep with bottle in his/her mouth? |
| | 3rd | 18–24 months | How much fruit juice does your child drink per day? |
| | | | Do you give vitamin supplements to your child? |
| | | | Does your child use a bottle to drink milk? |
| | 4th | 30–36 months | Does your child have a good appetite? |
| | | | How many meals does your child eat per day? |
| | 5th | 42–48 months | How much fruit juice does your child drink per day? |
| | | | How many meals does your child eat per day? |
| | 6th | 54–60 months | N/A |
| | 7th | 66–71 months | Does your child have breakfast every morning? |
| Oral health screening | 1st | 18–29 months | Visible dental plaque [a] |
| | | | Malocclusion state [a] |
| | | | Did your child stop using bottles for feeding? |
| | | | Does your child have remaining food particles in between teeth? |
| | | | How would you report your child's oral hygiene status? |
| | 2nd | 42–53 months | Visible dental plaque [a] |
| | | | Malocclusion state [a] |
| | | | Does your child have chewing problems? (not chewing or swallowing and pocketing food)? |
| | | | Does your child have remaining food particles in between teeth? |
| | | | How would you report your child's oral hygiene status? |
| | 3rd | 54–65 months | Visible dental plaque [a] |
| | | | Malocclusion state [a] |
| | | | Does your child have chewing problems? (not chewing or swallowing and pocketing food)? |
| | | | Does your child have remaining food particles in between teeth? |
| | | | How would you report your child's oral hygiene status? |

[a] Response provided by dentists responsible for oral health screening. NICU, Neonatal intensive care unit, N/A, not applicable

preterm birth and receiving an oral health screening. The regression model included 39,713 (47%) infants.

## Statistical analysis

SAS Enterprise Guide 7.1 (SAS Institute Inc., Cary, NC, United States) and R version 3.3.3 (R Foundation for Statistical Computing, Vienna, Austria) provided through the NHIS's server in South Korea were used for data analysis. The t-test was used for normally distributed continuous variables and the Chi-square test used for categorical variables. P-value less than 0.05 is considered as statistically significant. Multivariable logistical regression model and Wald test was used to evaluate an association between potential factors including preterm birth and completion of oral health screening. Birth history and demographic factors such as sex, disability and income were included to model and dietary habits were selected based on backward selection with at least marginal significance (p<0.2) on univariate analysis.

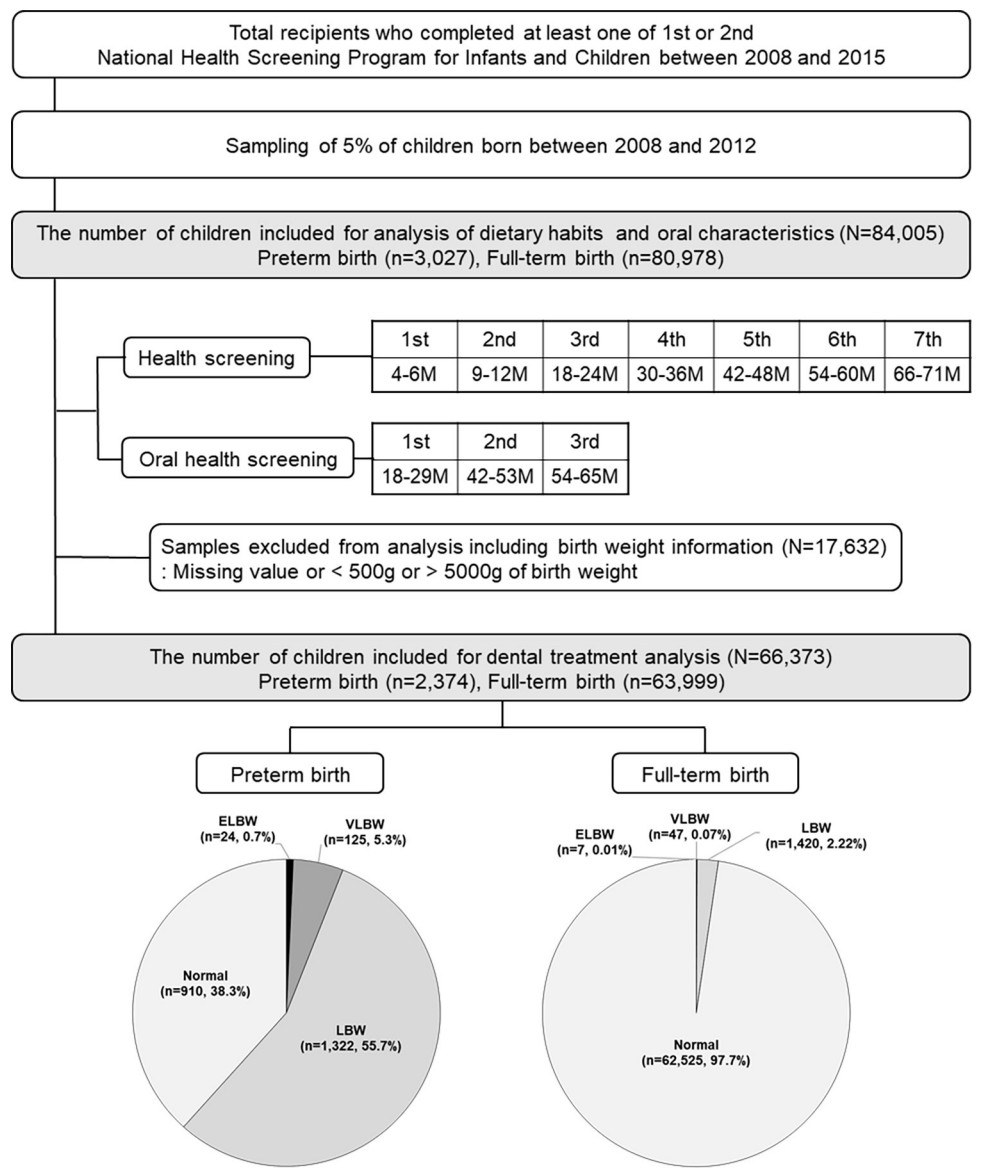

**Fig 1. Flow diagram of participant selection.** The proportion of preterm born infants were 3.6% (n = 2,374) of the study population. 38.3% of preterm children had normal birth weight, and 97.7% of the full-term children had normal birth weight (≥2500 g). LBW: low birth weight (≥1500 g and <2500 g), VLBW: very low birth weight (≥1000 g and <1500 g), ELBW: extremely low birth weight (≥500 g and <1000 g).

## Results

### Characteristics of study participants

Overall, the responses of survey and results of medical and oral screenings of 84,005 infants born between 2008 and 2012 who had completed at least the first or second health screening were analyzed (Fig 1). As a result, 96.4% (n = 80,978) of infants were born full-term and 3.6% (n = 3,027) were born preterm. The birth week (gestational age) was not recorded and therefore, birth weight distribution was utilized to evaluate the prematurity for dental treatment analysis (n = 66,373). In the full-term group, 97.7% (n = 62,525) of newborns had normal birth weight; in the preterm group, only 38.3% (n = 910) had normal birth weight. LBW was

**Table 2. Characteristics, medical history, and dietary habits of preterm and full-term children.**

| Properties | | Age at survey | Preterm birth, n (%) | Full-term birth, n (%) | p-value |
|---|---|---|---|---|---|
| Baseline Characteristics | Male | 4–6 months | 1,599 (52.8) | 41,744 (51.6) | 0.183 |
| | Low household income (<30 percentile) | 4–6 months | 483 (16.0) | 11,714 (14.5) | 0.022 [a] |
| | Disabled | 4–6 months | 2 (0.07) | 11 (0.01) | 0.023 [a] |
| Past medical history | ≥5 days NICU administration | 4–6 months | 1,047 (43.99) | 2,913 (4.50) | <0.001 [a] |
| | <2500 g birth weight | 4–6 months | 1,464 (61.67) | 1,474 (2.30) | <0.001 [a] |
| Dietary habits | Breast feeding | 4–6 months | 627 (26.28) | 28,047 (43.34) | <0.001 [a] |
| | Start of weaning food at > 6 months | 9–12 months | 641 (27.21) | 13,468 (21.50) | <0.001 [a] |
| | Night-time bottle feeding | 9–12 months | 273 (11.62) | 6,903 (11.03) | 0.394 |
| | ≥200ml sugar-added beverages | 18–24 months | 130 (5.91) | 4,520 (7.43) | 0.008 [a] |
| | Vitamin supplements intake | 18–24 months | 908 (41.16) | 24,580 (40.28) | 0.421 |
| | Bottle feeding | 18–24 months | 667 (30.24) | 14,776 (24.22) | <0.001 [a] |
| | Poor appetite | 30–36 months | 160 (7.06) | 3,050 (4.98) | <0.001 [a] |
| | ≥3 times of meals | 30–36 months | 2,021 (89.15) | 55,002 (89.86) | 0.284 |
| | ≥200ml sugar-added beverages | 42–48 months | 105 (7.24) | 2,522 (6.56) | 0.337 |
| | ≥3 times of meals | 42–48 months | 1,188 (81.71) | 31,387 (81.64) | 0.976 |
| | Irregular breakfast | 66–71 months | 201 (31.55) | 5,592 (31.83) | 0.920 |
| | Cessation of bottle feeding [b] | 18–29 months | 7 (38.89) | 351 (69.09) | 0.015 [a] |
| | Swallowing food without chewing [b] | 42–53 months | 96 (28.15) | 1,977 (22.71) | 0.023 [a] |
| | Swallowing food without chewing [b] | 54–65 months | 34 (22.97) | 704 (19.28) | 0.314 |

[a] chi-square test, p-value < 0.05

[b] Reported by dentists responsible for oral health screening, NICU, Neonatal intensive care unit

significantly higher in the preterm group (55.7%) than in the full-term group (2.2%). The evaluation at 4–6 months when the first health screening was performed, the preterm group had a significantly lower household income of less than 30 percentiles (p = 0.022) (Table 2). In addition, the proportion of children with disabilities in preterm birth was significantly higher (p = 0.023), however, the overall number of children certified disability registrations was lower. The proportions of VLBW and ELBW were also higher in the preterm group. NICU stay >5 days at the age of 4–6 months was significantly higher in the preterm group (44.0%) than in the full-term group (4.5%).

## Dietary habits of study participants

Table 2 presents the dietary habits of study participants. Full-term infants had a significantly higher percentage of breastfeeding (43.3%) than preterm infants (26.3%) (p<0.001). At 9–12 months, preterm babies started on weaning food significantly later than full-term babies (p<0.001); they also had poor appetite at 30–36 months (p<0.001) and swallowed food without chewing at 42–53 months (p = 0.023). Preterm babies showed a significantly higher proportion difference of about 6% in bottle feeding at 18–24 months (p<0.001), while the full-term babies showed a significantly higher proportion of cessation of bottle feeding at 18–29 months (p = 0.015). Preterm infants engaged in night-time bottle feeding for a longer duration and improper swallowing and chewing function than full-term infants.

## Oral characteristics of study participants

Parents of preterm group reported higher food impaction between teeth and poor oral hygiene at the age of 42–53 months however, only the latter showed the significant difference

($p$ = 0.046) (Table 3). Coincidentally, the presence of visible dental plaque was higher in the preterm group. Preterm infants showed higher rates of malocclusion than the full-term infants, but not at a significant level.

## Dental treatment characteristics of study participants

The percentage of visits to the dentist at least once were significantly higher in full-term group (75.8%) than preterm group (74.0%) ($p$ = 0.036) (Table 4). However, the average number of dental visits for children were not significant between full-term (5.94±5.43) and preterm groups (5.90±5.31) ($p$ = 0.783). Full-term children underwent 1 visit pulpectomy, pulpotomy and rubber dam application more often than the preterm children and required behavior management for less than 15 minutes. Full-term children had more glass ionomer restorations than preterm children ($p$ = 0.007) and preterm children had more sealants performed ($p$ = 0.034) when comparing dental treatment performed per dental visit. Table 5 lists comparison of number of dental treatments done according to oral health screening completion. There was no difference in dental treatments between full-term and preterm children if a child had never received oral health screening. For preterm children who have completed at least one oral health screening, 1-visit pulpectomy ($p$ = 0.007) and 2-visit pulpectomy ($p$ = 0.042) were performed less. Dietary habits and dental experiences of preterm children are summarized in Fig 2.

## Preterm birth and oral health screening completion

The multivariable logistic regression model included the potential variables shown in Table 6. The preterm birth group presented a significantly higher probability of oral health screening completion than the full-term birth group (aOR 1.141, 95% CI 1.015 to 1.283). Moreover, oral health screening completion was significantly higher in subjects who responded to breastfeeding at 4–6 months (aOR 1.094, 95% CI 1. 048 to 1.142) and vitamin supplements intake at 18–24 months (aOR 1.105, 95% CI 1.059 to 1.154). On the other hand, oral health screening completion was significantly lower in subjects who were male (aOR 0.956, 95% CI 0.917 to 0.997), had low household income (aOR 0.937, 95% CI 0.882 to 0.996), late start of weaning food in >6 months (aOR 0.901 95% CI 0.855 to 0.950), and breastfeeding at 18–24 months (aOR 0.846 95% CI 0.805 to 0.890).

**Table 3. Oral characteristics of preterm and full-term children in oral health screening.**

| Variables | Age at survey | Preterm birth, n (%) | Full-term birth, n (%) | *p*-value |
|---|---|---|---|---|
| Food impaction state between teeth | 18–29 months | 3 (16.67) | 148 (30.52) | 0.319 |
| | 42–53 months | 180 (52.63) | 4,166 (47.85) | 0.093 |
| | 54–65 months | 71 (47.97) | 1,691 (46.29) | 0.750 |
| Parent's report of child's poor oral hygiene | 18–29 months | 2 (11.11) | 54 (10.71) | 1.000 |
| | 42–53 months | 27 (7.89) | 459 (5.27) | 0.046 [a] |
| | 54–65 months | 5 (3.38) | 204 (5.59) | 0.331 |
| Visible dental plaque on teeth surface [b] | 18–29 months | 276 (33.78) | 8,265 (36.73) | 0.092 |
| | 42–53 months | 130 (38.01) | 3,304 (37.92) | 1.000 |
| | 54–65 months | 48 (32.43) | 1,288 (35.23) | 0.541 |
| Malocclusion state [b] | 18–29 months | 68 (8.32) | 1,548 (6.88) | 0.127 |
| | 42–53 months | 23 (6.73) | 381 (4.37) | 0.053 |
| | 54–65 months | 8 (5.41) | 143 (3.91) | 0.485 |

[a] chi-square test, *p*-value < 0.05

[b] Reported by dentists responsible for oral health screening

**Table 4. Comparison of dental treatments between preterm and full-term children.**

| | Dental experience | | | | | Average dental experience per 1 dental visit | | | | |
| --- | --- | --- | --- | --- | --- | --- | --- | --- | --- | --- |
| | Preterm birth (n = 2,374) | | Full-term birth (n = 63,999) | | *p*-value[b] | Preterm birth (n = 2,374) | | Full-term birth (n = 63,999) | | *p*-value[c] |
| | N | % | N | % | | Mean | SD | Mean | SD | |
| Dental visit | 1,756 | 74.0 | 48,540 | 75.8 | 0.036 [a] | 5.90 | 5.31 | 5.94 | 5.43 | 0.783[d] |
| Treatment | | | | | | | | | | |
| Periapical radiography | 1,003 | 42.2 | 27,560 | 43.1 | 0.432 | 0.41 | 0.26 | 0.42 | 0.26 | 0.308 |
| Extraction of primary tooth | 664 | 28.0 | 17,796 | 27.8 | 0.862 | 0.38 | 0.25 | 0.37 | 0.25 | 0.345 |
| Behavior management (≤15 min) | 195 | 8.2 | 6,190 | 9.7 | 0.018 [a] | 0.25 | 0.18 | 0.25 | 0.18 | 0.813 |
| Pulp treatment: 1-visit pulpectomy | 314 | 13.2 | 9,968 | 15.6 | 0.002 [a] | 0.34 | 0.28 | 0.35 | 0.28 | 0.476 |
| Pulp treatment: 2-visit pulpectomy | 313 | 13.2 | 9,360 | 14.6 | 0.051 | 0.41 | 0.34 | 0.42 | 0.36 | 0.858 |
| Pulp treatment: pulpotomy | 181 | 7.6 | 5,745 | 9.0 | 0.023 [a] | 0.17 | 0.14 | 0.19 | 0.14 | 0.207 |
| Glass ionomer restoration | 293 | 12.3 | 8,718 | 13.6 | 0.074 | 0.29 | 0.22 | 0.32 | 0.24 | 0.007 [a] |
| Rubber dam application | 409 | 17.2 | 12,809 | 20.0 | 0.001 [a] | 0.31 | 0.21 | 0.32 | 0.22 | 0.366 |
| Sealant | 212 | 8.9 | 5,715 | 8.9 | 0.999 | 0.21 | 0.15 | 0.19 | 0.15 | 0.034 [a] |
| Prophylaxis and scaling | 108 | 4.5 | 3,341 | 5.2 | 0.148 | 0.31 | 0.27 | 0.27 | 0.26 | 0.111 |

[a] *p*-value < 0.05
[b] chi-square test
[c] t-test
[d] Average number of dental visits

## Discussion

In this study, the association between dietary habits and oral characteristics of preterm children was evaluated using a nationwide database. Full-term babies usually had normal weight at birth, and preterm babies usually had low body weight. Duration of NICU stay was longer,

**Table 5. Comparison of dental treatments according to oral health screening completion.**

| | Oral health screening completed [c] (n = 39,248) | | | | | Oral health screening incompleted (n = 27,125) | | | | |
| --- | --- | --- | --- | --- | --- | --- | --- | --- | --- | --- |
| | Preterm birth (n = 1,416) | | Full-term birth (n = 37,832) | | *p*-value [b] | Preterm birth (n = 958) | | Full-term birth (n = 26,167) | | *p*-value [b] |
| | N | % | N | % | | N | % | N | % | |
| Dental visit | 1,092 | 77.1 | 29,847 | 78.9 | 0.108 | 664 | 69.3 | 18,693 | 71.4 | 0.153 |
| Treatment | | | | | | | | | | |
| Periapical radiography | 621 | 43.9 | 17,049 | 45.1 | 0.369 | 382 | 39.9 | 10,511 | 40.2 | 0.855 |
| Extraction of primary tooth | 404 | 28.5 | 10,697 | 28.3 | 0.834 | 260 | 27.1 | 7,099 | 27.1 | 0.994 |
| Behavior management (≤15 min) | 111 | 7.8 | 3,630 | 9.6 | 0.027 [a] | 84 | 8.8 | 2,560 | 9.8 | 0.298 |
| Pulp treatment: 1-visit pulpectomy | 182 | 12.9 | 5,863 | 15.5 | 0.007 [a] | 132 | 13.8 | 4,105 | 15.7 | 0.110 |
| Pulp treatment: 2-visit pulpectomy | 172 | 12.1 | 5,316 | 14.1 | 0.042 [a] | 141 | 14.7 | 4,044 | 15.5 | 0.535 |
| Pulp treatment: pulpotomy | 96 | 6.8 | 3,097 | 8.2 | 0.057 | 85 | 8.9 | 2,648 | 10.1 | 0.208 |
| Glass ionomer restoration | 176 | 12.4 | 5,205 | 13.8 | 0.153 | 117 | 12.2 | 3,513 | 13.4 | 0.279 |
| Rubber dam application | 238 | 16.8 | 7,779 | 20.6 | 0.001 [a] | 171 | 17.8 | 5,030 | 19.2 | 0.289 |
| Sealant | 121 | 8.5 | 3,410 | 9.0 | 0.545 | 91 | 9.5 | 2,305 | 8.8 | 0.460 |
| Prophylaxis and scaling | 77 | 5.4 | 2,287 | 6.0 | 0.346 | 31 | 3.2 | 1,054 | 4.0 | 0.219 |

[a] *p*-value < 0.05
[b] chi-square test
[c] Infants who have completed at least one oral health screening

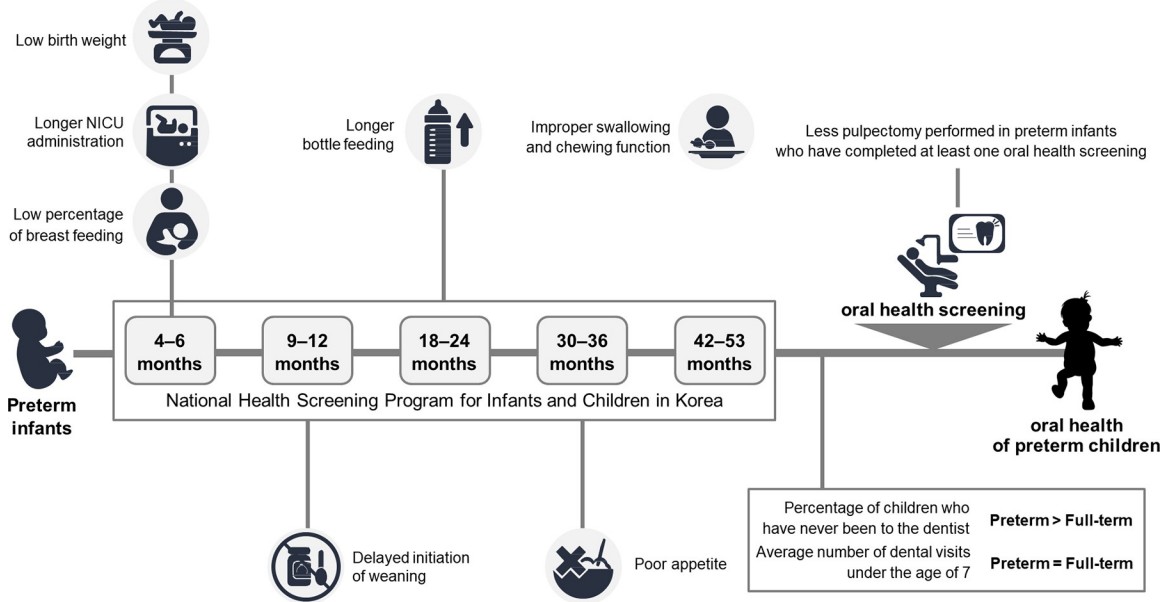

**Fig 2. Summary of dietary habits, oral characteristics, and dental treatments in preterm infants in National Health Screening Program for Infants and Children in Korea, NICU; neonatal intensive care unit administration.**

start of breastfeeding was more delayed, period of bottle feeding was longer, and incidence of swallowing food without chewing was higher in the preterm group than in the full-term group. At all stages of oral health screening, preterm infants showed malocclusion but few visits to the dental office and, therefore, few dental treatment experiences.

Preterm babies had significantly lower sugar intake at 18–24 months but higher levels of sugar intake and vitamin supplementation after 42 months than full-term babies. The WHO recommends exclusive breastfeeding in the first 6 months of life and complementary feeding along with breast milk for 2 years [24]. Premature infants admitted to the NICU right after birth, unfortunately, are separated from their mothers and are unable to receive sufficient or any breast milk [25]. Premature infants are usually under gavage feeding for a long time, and this leads to a high risk of failure of oral feeding [26]. Oral feeding dysfunction can delay discharge from the NICU and postpone the start of weaning food, often to after 6 months of age [27], and premature infants may not be ready to start eating solid foods at that age.

The prevalence of night-time bottle feeding was similar for both groups at 4–6 months, but preterm infants were significantly prone to bottle feeding at night time until 18–24 months. The guidelines of the American Association of Pediatric Dentistry suggest that bottle feeding should stop between the ages of 12 and 18 months and drinking from a cup from should be encouraged from the first birthday onward [28]; it is also strongly recommended that infants should not be put to sleep with a formula bottle. Long-term bottle feeding leads to a reduction in masseter muscle activity [29], posterior crossbite, and altered occlusion, and these features are more pronounced in infants with non-nutritive sucking habit [30]. Bottle feeding also has negative consequences on orofacial development, such as lack of lip seal, persistent infantile swallowing, and reduced nasal breathing [31].

It is interesting to note that similar contextual questions such as day-time and night-time bottle feeding were asked in different sessions of the medical and oral screening. The oral screening session included a question on bottle feeding up to 18–24 months, while the health screening session included the same question for 9–12 months. In the health screening session,

**Table 6. Multivariable logistic regression analysis for oral health screening completion.**

| Variable and classification | aOR | 95% CI of aOR | | p-value |
|---|---|---|---|---|
| | | Lower | Upper | |
| Birth history | | | | |
| Full-term birth (Ref.) | | | | |
| Preterm birth | 1.141 | 1.015 | 1.283 | 0.0269[a] |
| Sex | | | | |
| Female (Ref.) | | | | |
| Male | 0.956 | 0.917 | 0.997 | 0.0366* |
| Disabled | | | | |
| No (Ref.) | | | | |
| Yes | 0.754 | 0.125 | 4.541 | 0.7582 |
| Low household income (<30 percentile) | | | | |
| No (Ref.) | | | | |
| Yes | 0.937 | 0.882 | 0.996 | 0.0373[a] |
| Breast feeding at 4–6 months | | | | |
| No (Ref.) | | | | |
| Yes | 1.094 | 1.048 | 1.142 | <0.001[a] |
| Start of weaning food | | | | |
| ≤6 months (Ref.) | | | | |
| >6 months | 0.901 | 0.855 | 0.950 | 0.0001[a] |
| sugar-added beverages at 18–24 months | | | | |
| < 200ml (Ref.) | | | | |
| ≥200ml | 0.954 | 0.880 | 1.035 | 0.2560 |
| Vitamin supplements intake at 18–24 months | | | | |
| No (Ref.) | | | | |
| Yes | 1.105 | 1.059 | 1.154 | <0.001[a] |
| Breast feeding at 18–24 months | | | | |
| No (Ref.) | | | | |
| Yes | 0.846 | 0.805 | 0.890 | <0.001[a] |

aOR, adjusted Odds Ratio

CI, Confidence Interval

Ref., Reference group

[a]Multivariable logistical regression model and Wald test, p-value < 0.05.

closed questions on dietary habits were asked, such as three or more meal times, poor appetite, and intake of vitamin supplements. However, in the dental screening, emphasis was placed on questions related to oral function, such as swallowing and chewing. Dentists tend to focus on oral functions that are often overlooked by parents or physicians.

A previous systematic review and meta-analysis showed that bottle-fed children had more dental caries than breast milk-fed children [32]. In another study, preterm babies had not only poor appetite and low motivation to eat but also high levels of neophobia and pickiness, resulting in difficult eating behaviors and slow and long feeding processes [33]. Children who do not have breakfast daily or eat fewer than five servings of fruit and vegetables per day showed higher odds of experiencing caries in primary teeth [34]. At 42–53 months, preterm children notably swallowed food without chewing. Dysphagia and aspiration are major challenges for preterm babies as they have poor timing of swallowing and poor coordination of the sucking, swallowing, and breathing actions [35]. Simultaneous and sequential muscle activations, involving multifunctional neural and mechanical circuits, are required to maintain effective

patterns of oral feeding. The impaired interaction between the neurological system and muscle tones in preterm babies results in an unstable suck-swallow-breathe pattern then in turn causes rapid exhaustion and inadequate consumption during oral feeding [21].

Preterm babies showed a higher probability of developing malocclusion at all stages of oral health screenings. Prematurely born children are more susceptible to malocclusion traits such as dental crowding [36], bilateral crossbite [37], and deep bite [38] than full-term children. Premature birth is also associated with skeletal defects such as palatal grooving, asymmetrical jaw, and high-arched palate [39, 40]. Further, 42–53-month-old preterm infants had increased proportions of visible dental plaque and food impaction on the entire dentition. Parents of preterm infants were more aware of their children's poor oral hygiene than parents of full-term infants. It is difficult to observe caries at 18–29 months when primary molars start to erupt but have not fully erupted to the occlusal surface. Child's oral hygiene becomes more noticeable to parents of preterm infants when tooth cavitation due to dental caries is recognized after the complete eruption of primary dentition. This is the underlying reason why preterm parents of 42–53 months significantly recognized the child's oral hygiene to be poor. However, there was no significant difference between the two groups at the follow-up examination at 54–65 months. It was difficult to examine the precise state of oral health because the rate of oral health screening was lower than that of health screening. Furthermore, few preterm infants underwent oral screening, possibly reflecting ignorance of the importance of regular dental visits. According to data in the 2018 National Dental Screening Statistical Yearbook, provided by the NHIS, rates of health screening and dental screening for infants were 74.5% and 45.2%. The first, second, and third oral screening rates showed a decreasing trend, at 56.8%, 44.6%, and 34.9%, respectively [41].

This study found that the dietary habits and oral hygiene of preterm infants were predisposing factors for dental health problems in early childhood. However, controversially, rates of dental treatments such as behavior management, pulp treatment, or glass ionomer restoration were lower in preterm infants than in full-term infants. There were more preterm children who have never visited dental office. However, the average dental visit was similar between full-term and preterm children. For preterm born child who received at least one oral health screening service, the number of pulpectomy aiming to treat advanced dental caries that have invaded the dental pulp decreased. Therefore, this finding can be attributed to the fact that should preterm children make the effort to complete oral screening or visit dental office for oral examination, preventive and prophylactic measures with oral hygiene instructions may be applied and ultimately require less dental treatments in the future. In a case-control observational study, the prevalence of caries was higher in preterm infants (50.5%) than in full-term infants (12.5%) (Decayed, Missing and Filled Teeth Index (DFMT) 1.0 for preterm infants, DMFT 0.3 for full-term infants) [18]; in addition, the risk of developing caries, initial lesions, and gingivitis was higher in preterm infants and even more pronounced in the extremely preterm subgroup than in full-term infants. However, a few studies showed contrasting results. One study that examined the saliva of children born with very low or low body weight showed no statistically significant difference in the prevalence of *S. mutans* between these groups [42] and that parents of preterm infants were more meticulous in oral hygiene practices than those of full-term infants. However, very-low- and low-birth-weight children showed higher consumption of sweetened drinks during the day and night, being highly susceptible of dental caries. The available data from this study, unfortunately, could not provide index such as DMFT to accurately and objectively represent oral health. However, the regression model supports higher competence of oral health screening in preterm born group, more significantly higher if one was breastfeeding at 4–6 months and taking vitamin supplements at 18–24 months. On the contrary, preterm born children who started weaning food later than 6 months or were

breastfeeding at 18–24 months had lower completion which can be attributed to the fact that, parents are less aware of the need of oral health screening because an obvious problem related to oral health is not yet visible.

With notable advances in neonatal care, the survival of preterm infants has increased, and many studies have focused on the prevalence of a broad range of disabilities, neurological or developmental impairments, and medical conditions [43]. Researchers at Nottingham, London, Sheffield, and Oxford found that 49% of preterm babies had some form of disability, most commonly with non-fluent gait, impaired sight and hearing, delay in talking, and inability to eat with both hands [44]. Preterm infants need early interventions and special care in the early years of life to prevent subsequent dental health problems. Pediatric dentists in collaboration with appropriate medical health care professionals should carry out careful evaluations and focus on prevention of dental problems and treatment planning.

This study has some limitations. The study data included 5% of infants who had completed medical and dental screening examinations. This data does not provide personal information but are only classified as person identification number and therefore impossible to follow-up the patient during the screening period. The dates of medical and oral screening sessions do not match and therefore is not easy to study data collected at different points in time. The inability to compare the results of the same period is another limitation. According to birth rate by gestational age provided by the Korean Statistical Information Service (KOSIS), 5.5% of babies were born before 37 weeks of pregnancy in 2008 [45], a value higher than that reported in this study. In 2020, the preterm birth rate in South Korea was 8.4%, which was lower than the global rate of 12% [24]. A large proportion of health and oral health screening data depends on answers to screening questionnaires completed by parents. Parents of preterm children were asked to provide the gestational age at birth; therefore, the risk of recall bias cannot be ignored. This is why there may be differences between data provided by the KOSIS and NHIS.

The health screening rate was low and the oral screening rate was even lower, usually lower than 50%; therefore, data must be interpreted carefully. Treatment records showed only data of reimbursed treatments; however, in pediatric dental management, there are many more non-reimbursed treatments. Thus, the health records of medical and dental history did not reflect the exact treatments received by the participants. In addition, very few extremely preterm infants were included in this study; therefore, the prevalence of oral health problems may have been underestimated. We found that oral health was affected by gestational age and weight at birth; however, only 0.2% of the study sample had low birth weight below 1500g, and 0.04% had very low birth weight below 1000g.

## Conclusion

Preterm group showed a higher percentage of absence of dental visit even though dietary habits such as longer bottle feeding, poor appetite, and improper swallowing and chewing function were evident to worsen oral condition. However, for preterm infants who visited dental offices for oral examination, there was no difference in the average number of dental visits. 1 and 2 visit pulpectomy significantly decreased in preterm infants who completed at least one oral health screening. The NHSIC can be an effective policy for oral health management in preterm born babies and raising the awareness of oral health policy for preterm infants is needed in the future.

## Acknowledgments

The authors are especially grateful to Ji Young Park for assisting raw data collection from NHIS. Data collection from NHIS has many complicated steps of verifications and not easily assessed even for professionals.

## Author Contributions

**Conceptualization:** Ko Eun Lee, Jeong Eun Shin, Hoi-In Jung, Chung-Min Kang.

**Data curation:** Lan Herr, Juhyun Chung, Ko Eun Lee, Jung Ho Han, Jeong Eun Shin, Hoi-In Jung, Chung-Min Kang.

**Formal analysis:** Juhyun Chung, Hoi-In Jung, Chung-Min Kang.

**Funding acquisition:** Chung-Min Kang.

**Investigation:** Lan Herr, Juhyun Chung, Hoi-In Jung, Chung-Min Kang.

**Methodology:** Juhyun Chung, Hoi-In Jung.

**Project administration:** Chung-Min Kang.

**Resources:** Hoi-In Jung, Chung-Min Kang.

**Software:** Hoi-In Jung.

**Supervision:** Ko Eun Lee, Jung Ho Han, Jeong Eun Shin, Hoi-In Jung, Chung-Min Kang.

**Validation:** Chung-Min Kang.

**Visualization:** Chung-Min Kang.

**Writing – original draft:** Lan Herr, Juhyun Chung.

**Writing – review & editing:** Chung-Min Kang.

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
