## [Decision Letter · Decision Letter 0]

12 Sep 2022

PONE-D-22-19700Oral characteristics and dietary habits of preterm children: A retrospective study using a nationwide infant medical and oral screening cohortPLOS ONE

Dear Dr. Kang,

Thank you for submitting your manuscript to PLOS ONE. After careful consideration, we feel that it has merit but does not fully meet PLOS ONE’s publication criteria as it currently stands. Therefore, we invite you to submit a revised version of the manuscript that addresses the points raised during the review process.

We look forward to receiving your revised manuscript.

Kind regards,

Kuo-Cherh Huang

Academic Editor

PLOS ONE

Journal Requirements:

"We are grateful to Ji Young Park for assistance for data collection. This study used NHIS-INCHS data (NHIS-2021-2-104) made by National Health Insurance Service (NHIS). The authors declare no conflict of interest with the NHIS."

"This work was supported by the National Research Foundation of Korea(NRF) grant funded by the Korea government(MSIT) (No. NRF-2020R1G1A1100275) award to CMK. The funders had no role in study design, data collection and analysis, decision to publish, or preparation of the manuscript."

Additional Editor Comments:

Dear Dr. Kang,

We appreciate your submission to PLoS ONE. Although your paper is interesting, all three reviewers have provided a variety of important concerns, notably the research methodology and data analyses as well as presentations of analytical results of your study. Here, I would like to bring up a couple of critical points:

From Reviewer 1: “In addition, if it’s a true longitudinal study, you should follow-up the patient during the observation period to observe oral health and changes in lifestyle that can affect it.”; “The survey was only about the ratio and type of premature birth, analysis, and questionnaire, but I don’t think there’s any strong correlation, and there is lack of explanatory power to assess the results from citing other studies.” It is quite evident that you need to defend your research methodology and statistical techniques; otherwise, you should revise them, accordingly.

From Reviewer 2: “However, the authors merely presented the limited findings by a simple analytical fashion. In this retrospective study, readers may be interested in the differences in the potential factors for the poor oral health (as well as dietary habits or fewer dental treatment experiences) between the preterm birth and the full-term birth groups. The authors should perform the relevant regression analysis to explore them.” By the same token, Reviewer 2 had grave concerns with respect to your statistical analyses as Reviewer 1 held.

From Reviewer 3: “The current table and picture are not enough to show the results and conclusions. For example, the percentages listed in Figure 2 are intuitively difficult to understand.”; “There are some parts in the text that don't match the percentage. In a study using big data, accurate arithmetic expression without errors is essential to assert the characteristics of groups through representative values and present them as conclusions.” As Reviewer 3 　sharply pointed out, you need to be more careful about the validity and accuracy of statistics in your work.

Please respond to each comment of the reviewers carefully and thoroughly. Please explain where you feel you cannot completely agree with reviewers’ suggestions.

Kuo-Cherh Huang

Reviewers' comments:

Reviewer's Responses to Questions

**Comments to the Author**

1. Is the manuscript technically sound, and do the data support the conclusions?

Reviewer #1: Partly

Reviewer #2: Partly

Reviewer #3: Partly

2. Has the statistical analysis been performed appropriately and rigorously? 

Reviewer #1: Yes

Reviewer #2: No

Reviewer #3: Yes

3. Have the authors made all data underlying the findings in their manuscript fully available?

Reviewer #1: Yes

Reviewer #2: No

Reviewer #3: Yes

4. Is the manuscript presented in an intelligible fashion and written in standard English?

Reviewer #1: Yes

Reviewer #2: Yes

Reviewer #3: Yes

5. Review Comments to the Author

Reviewer #1: The manuscript is very interesting as the first national study to investigate the current status of premature babies and the lifestyle related to oral health. However, there are some shortcomings that require improvement. In the research method, it is necessary to explain more accurately what items were investigated in this study. Currently, there are only survey contents, and it is necessary to describe what content was extracted as a key element and compared based on the answers of the survey.

In addition, if it’s a true longitudinal study, you should follow-up the patient during the observation period to observe oral health and changes in lifestyle that can affect it. Alternatively, different samples should be investigated on the premise that the population is the same at different points in time so that they can know the change from year to year. In the current study, it is difficult to say that it is a true longitudinal study though it has period to increase the number of samples.

The only difference between the experimental group and the control group is the difference in perception of the parents' oral health at 42-53 months. There is a slight statistically significant difference. Why? should be dealt with in more detail in the consideration.

If the babies come to the dentist more often, more treatment will be done, and if there is no significant difference in oral condition, you may even conclude that full-term children's oral health is worse than preterm children. Since the questionnaire items of national screening are too simple, it is difficult to compare the information of the two groups. All explanations are not based on the results obtained in the current study, but are all guessed using previous studies. Changes in oral conditions and oral health habits according to age changes have not been studied.

Now, I will ask the author who submitted the manuscript.

First, why didn't you use the latest data?

Second, why did you not attempt to interpret it in connection with oral condition data such as dmfs?

Third, why didn't you study the data (year) collected at different points in time?

The survey was only about the ratio and type of premature birth, analysis, and questionnaire, but I don’t think there’s any strong correlation, and there is lack of explanatory power to assess the results from citing other studies. Although it has the advantage of being the first nationwide research, it is not suitable for publication due to insufficient research contents. Please reinforce the contents after a major revision and submit it again.

Reviewer #2: (1) The authors concluded that preterm infants had poor oral health and dietary habits and fewer dental treatment experiences than full-term infants. There were many valuable variables collected in the nationwide cohort study. However, the authors merely presented the limited findings by a simple analytical fashion. In this retrospective study, readers may be interested in the differences in the potential factors for the poor oral health (as well as dietary habits or fewer dental treatment experiences) between the preterm birth and the full-term birth groups. The authors should perform the relevant regression analysis to explore them.

(2) In this study, data on socio-economic variables, disability, and medical resource utilization status such as consultations and medical screening were collected; please show the information, which giving readers a clear feature regarding the study population.

(3) Please place the numbers of the preterm birth and the full-term birth groups respectively in Tables 2, 3 and 4, which can make the relevant information clear and easy for readers to understand.

Reviewer #3: 1. The author's study shows that the oral hygiene-related characteristics of preterm baby are very bad and dangerous compared to full-term baby, and it would be better to show it in an arithmetically impactful schematic that is easier to understand.

2. The current table and picture are not enough to show the results and conclusions. For example, the percentages listed in Figure 2 are intuitively difficult to understand.

3. There are some parts in the text that don't match the percentage. In a study using big data, accurate arithmetic expression without errors is essential to assert the characteristics of groups through representative values and present them as conclusions.

4. The comparison of oral characteristics of preterm baby and full term baby shown in Table 3 shows no significant difference in all cases except for '42-53 months of oral hygiene'. Considering the tone of the text that Preterm baby is not better, Table 3 seems to show the opposite results.

5. In addition, "food impaction", "poor oral hygiene", and "visible dental plaque" are all overlapping items, but they all marked differently and processed statistics separately, so please explain why they were classified separately.

6. "Parent-reported poor oral hygiene and food impaction between teeth (Table 3) at the age of 42–53 months was higher in the preterm group than in the full-term group (p < 0.05)."

=> In addition, according to Table 3 in this sentence, 'food impaction' does not show a significant difference, but it is stated that it shows a significant difference together.

7. The results in Table 4 show that the full-term birth group as a whole received more dental treatment in most items than the preterm birth group. However, from these results, it may be interpreted that the full-term birth group is good at dental care, but it may be interpreted that the full-term birth group had more cavities and received more treatment. Therefore, it seems that the clinical data should be further supplemented for accurate analysis of Table 4.

6. PLOS authors have the option to publish the peer review history of their article (what does this mean?). If published, this will include your full peer review and any attached files.

Reviewer #1: No

Reviewer #2: No

Reviewer #3: No

---

## [Author Response · Author response to Decision Letter 0]

1 Dec 2022

Dear Editor and Reviewers, 

We thank you for giving us the opportunity to submit a revised draft of manuscript and your thoughtful suggestions and insights. The manuscript has benefited from these insightful suggestions. I look forward to working with you and the reviewers to move this manuscript closer to publication in PLOS ONE. The manuscript has been rechecked and the necessary changes have been made in accordance with the reviewers’ suggestions. 

Thank you for your consideration. I look forward to hearing from you.

Here is a point-by-point response to the reviewers’ comments and concerns. The reviewers’ comments are in bold. Our response follows and edited manuscript is highlighted in yellow.

Additional Editor Comments:

Thank you for your comment. We looked over the entire manuscript to ensure it meets PLOS ONE’s style requirements including file naming. 

2. Please remove any funding-related text from the manuscript and let us know how you would like to update your Funding Statement. We note that you have provided funding information that is not currently declared in your Funding Statement. However, funding information should not appear in the Acknowledgments section or other areas of your manuscript. Please remove any funding-related text from the manuscript and let us know how you would like to update your Funding Statement. Please include your amended statements within your cover letter; we will change the online submission form on your behalf.

Thank you for your comment. We reorganized Acknowledgements, Funding and Data Availability section for more clarification-which is also related to your next comment (comment #3). 

Acknowledgement

This part was originally written to pay special gratitude to Ji Young Park who helped the authors with raw data extraction from NHIS. NHIS information is private and not easily assessed and has many complicated steps of verifications for extracting data source, even for professionals. However, this is not related to funding. We have edited Acknowledgement section to be more specific and removed source of confusion.

The authors are especially grateful to Ji Young Park for assisting raw data collection from NHIS. Data collection from NHIS has many complicated steps of verifications and not easily assessed even for professionals. 

Funding

All funding related information regarding this manuscript is written below. 

This work was supported by the National Research Foundation of Korea(NRF) grant funded by the Korea government(MSIT) (No. NRF-2020R1G1A1100275) awarded to Chung-Min Kang. The funders had no role in study design, data collection and analysis, decision to publish, or preparation of the manuscript.

Data availability statement

The minimal data set underlying the results described in our manuscript was collected from data provided by National Health Screening Program for Infants and Children (NHSIC) of the National Health Insurance Service (NHIS) of Korea. This data cannot be shared with the public and only the verified researchers who meet the accessibility criteria has access in only permitted time. The authors had no privileges related to these criteria compared to others. The authors declare no conflict of interest with NHIS.

3. In your Data Availability statement, you have not specified where the minimal data set underlying the results described in your manuscript can be found.

Thank you for your thoughtful comment. We have edited and specified where the minimal data set underlying the results can be found in Data availability statement. Please refer to the previous comment for revised text. 

4. From Reviewer 1: It is quite evident that you need to defend your research methodology and statistical techniques; otherwise, you should revise them, accordingly.

From Reviewer 2: By the same token, Reviewer 2 had grave concerns with respect to your statistical analyses as Reviewer 1 held.

From Reviewer 3: As Reviewer 3 sharply pointed out, you need to be more careful about the validity and accuracy of statistics in your work.

Thank you for your comment. We revised table 4 and added table 5 to improve comparing the dental experience of preterm birth and full-term birth. The average dental experience per 1 dental visit, the sum of each dental treatment per single dental visit, are added as the outcome variable. The dental experience and the average dental experience per 1 dental visit of those who had undergone an oral health screening at least once and those who had never had oral health screening are analyzed separately. By making these changes, we hope we have provided more clear research methodology and valid statistics. Reviewer 1,2 and 3 have commonly given comments about our methodology and statistics. We have edited our manuscript and written our responses to each reviewer in hopes of elucidating more precisely study design and results of our study.

---

## [Decision Letter · Decision Letter 1]

19 Dec 2022

PONE-D-22-19700R1Oral characteristics and dietary habits of preterm children: A retrospective study using National Health Screening Program for Infants and ChildrenPLOS ONE

Dear Dr. Kang,

Thank you for submitting your manuscript to PLOS ONE. After careful consideration, we feel that it has merit but does not fully meet PLOS ONE’s publication criteria as it currently stands. Therefore, we invite you to submit a revised version of the manuscript that addresses the points raised during the review process.

We look forward to receiving your revised manuscript.

Kind regards,

Kuo-Cherh Huang

Academic Editor

PLOS ONE

Journal Requirements:

Additional Editor Comments :

Dear Dr. Kang,

Thank you for submitting your revised manuscript to PLoS ONE. I have carefully read your revised manuscript and responses to the previous round of review comments. I appreciate your efforts. Although Reviewer 1 recommended favorably regarding your revised work, Reviewer 2 still held strong opinions against your paper. The critical concern of Reviewer 2 was related to your statistical analysis: “The authors still presented the limited findings by a simple analytical fashion in the revised manuscript. In this retrospective study, readers may be interested in the differences in the potential factors for the poor oral health (as well as dietary habits or fewer dental treatment experiences) between the preterm birth and the full-term birth groups. The authors can use the relevant regression analysis to explore them in the preterm birth and the full-term birth groups, respectively. Why did the authors NOT perform these regression analyses?” Actually, a similar issue was raised by Reviewer 1 in his/her previous review comments: “… but I don’t think there’s any strong correlation, and there is lack of explanatory power to assess the results from citing other studies. Although it has the advantage of being the first nationwide research, it is not suitable for publication due to insufficient research contents”.. Even though in your rebuttal letter you had explained why you opted not to carry out regression modeling, it is quite obvious that Reviewer 2 was not convinced at all by your rationales.

One criterion for publication at PLoS ONE is: ”Experiments, statistics, and other analyses are performed to a high technical standard and are described in sufficient detail.” I think it would be better to present the statistics (results of regression analysis) and let the reader read and make their own judgment in respect of the study power of your work, instead of simply arguing that “we did not find too many outcomes with significant difference from this study and thus decided regression model is not fit for this manuscript.” In fact, it is kind of surprising to me with reference to the declared outcomes, considering it is a nationwide, population-based longitudinal study and the sample size is sufficiently large (N = 84,005).

Kuo-Cherh Huang

Reviewers' comments:

Reviewer's Responses to Questions

**Comments to the Author**

1. If the authors have adequately addressed your comments raised in a previous round of review and you feel that this manuscript is now acceptable for publication, you may indicate that here to bypass the “Comments to the Author” section, enter your conflict of interest statement in the “Confidential to Editor” section, and submit your "Accept" recommendation.

Reviewer #1: All comments have been addressed

Reviewer #2: (No Response)

2. Is the manuscript technically sound, and do the data support the conclusions?

Reviewer #1: Yes

Reviewer #2: No

3. Has the statistical analysis been performed appropriately and rigorously? 

Reviewer #1: Yes

Reviewer #2: No

4. Have the authors made all data underlying the findings in their manuscript fully available?

Reviewer #1: Yes

Reviewer #2: No

5. Is the manuscript presented in an intelligible fashion and written in standard English?

Reviewer #1: Yes

Reviewer #2: No

6. Review Comments to the Author

Reviewer #1: I am pleased to be in charge of the review of the revised paper. I have confirmed that very sincere responses and corrections have been made to the points pointed out by several reviewers. Also I confirmed that it was reflected in the revised manuscript. Despite the limitations of the study, this study contains meaningful data to publish and examines the relationship between low birth weight and oral health. Therefore, it is judged to be suitable for publication.

Reviewer #2: The authors still presented the limited findings by a simple analytical fashion in the revised manuscript. In this retrospective study, readers may be interested in the differences in the potential factors for the poor oral health (as well as dietary habits or fewer dental treatment experiences) between the preterm birth and the full-term birth groups. The authors can use the relevant regression analysis to explore them in the preterm birth and the full-term birth groups, respectively. Why did the authors NOT perform these regression analyses? Please explain it.

7. PLOS authors have the option to publish the peer review history of their article (what does this mean?). If published, this will include your full peer review and any attached files.

Reviewer #1: No

Reviewer #2: No

---

## [Author Response · Author response to Decision Letter 1]

31 Jan 2023

Dear Editor and Reviewers, 

We thank you for giving us the opportunity to make a minor revision. The authors tried to answer the reviewer's additional requests as sincerely as possible. I look forward to working with you and the reviewers to move this manuscript closer to publication in PLOS ONE. Thank you for your consideration. I look forward to hearing from you. Here is a point-by-point response to the reviewers’ comments and concerns. The reviewers’ comments are in bold. Our response follows and edited manuscript is highlighted in yellow.

Editor Comments: 

I think it would be better to present the statistics (results of regression analysis) and let the reader read and make their own judgment in respect of the study power of your work, instead of simply arguing that “we did not find too many outcomes with significant difference from this study and thus decided regression model is not fit for this manuscript.” In fact, it is kind of surprising to me with reference to the declared outcomes, considering it is a nationwide, population-based longitudinal study and the sample size is sufficiently large (N = 84,005).

Our response: Thank you for your thoughtful comment. Like you and reviewer 2 suggested, we have included results of regression analysis in our revised manuscript and incorporated our thoughts and interpretation in the discussion as well. We fully respect the statistics criteria for publication at Plos ONE and we have put in effort to meet the standards. Once again, thank you for your advice and comments. 

Reply to reviewer #1 

I am pleased to be in charge of the review of the revised paper. I have confirmed that very sincere responses and corrections have been made to the points pointed out by several reviewers. Also I confirmed that it was reflected in the revised manuscript. Despite the limitations of the study, this study contains meaningful data to publish and examines the relationship between low birth weight and oral health. Therefore, it is judged to be suitable for publication.

Our response: Thanks for your response. All authors are delighted and honored that the revised manuscript can satisfy your request.

Reply to reviewer #2

The authors still presented the limited findings by a simple analytical fashion in the revised manuscript. In this retrospective study, readers may be interested in the differences in the potential factors for the poor oral health (as well as dietary habits or fewer dental treatment experiences) between the preterm birth and the full-term birth groups. The authors can use the relevant regression analysis to explore them in the preterm birth and the full-term birth groups, respectively. Why did the authors NOT perform these regression analyses? Please explain it.

Our response: Thank you for your valuable advice. We have constructed multivariable logistic regression on the association between preterm birth and receiving oral health screening. The model includes characteristics of study participants, medical history, and dietary habits as explanatory variables. 

There is no absolute and objective factor like DMFT to represent oral health for comparison in the available study data. In our results, we have found that preterm born babies do not necessarily correlate to having more dental treatments, but actually, less or similar dental treatment experiences compared to full-term born groups and therefore it is difficult to conclude preterm birth is a predisposing factor for poor oral health. 

In this study, the authors would like emphasize that preterm born children who have completed oral health screening have decreased dental treatment experiences, especially the need of pulpectomy which is done to treat severe dental caries or caries with lesion extended to periapical region. Therefore, the authors have included factors that could potentially affect oral health screening in the regression model. 

The regression model depicted that the rate of completing oral health screening is higher in preterm born children than full-term born children which ultimately suggests that even though preterm born children have long bottle feeding period, low appetite and poor oral and dietary habits, those who put in the effort to complete oral health screening, can potentially attenuate the possibility of severe dental treatments through regular oral examination and oral health instructions.

---

## [Editor Report · Decision Letter 2]

3 Feb 2023

Oral characteristics and dietary habits of preterm children: A retrospective study using National Health Screening Program for Infants and Children

PONE-D-22-19700R2

Dear Dr. Kang,

We’re pleased to inform you that your manuscript has been judged scientifically suitable for publication and will be formally accepted for publication once it meets all outstanding technical requirements.

Kind regards,

Kuo-Cherh Huang

Academic Editor

PLOS ONE

Additional Editor Comments (optional):

Dear Dr. Kang,

I appreciate your additional work of carrying out multiple variable analysis as requested by the reviewer. I only have some remaining relatively minor suggestions: for the newly added Table 6, since you included an exact p-value column within the table along with a table title indicating “Multivariable logistic regression analysis”, the footnote became redundant, then -- “* Multivariable logistical regression model and Wald test, p<0.05”. On the other hand, Tables 2 and 3 need to add footnotes to be explicit about the analysis approach. Thank you.

Kuo-Cherh Huang
---

## [Editor Report · Acceptance letter]

8 Feb 2023

PONE-D-22-19700R2 

Oral characteristics and dietary habits of preterm children: A retrospective study using National Health Screening Program for Infants and Children 

Dear Dr. Kang:

I'm pleased to inform you that your manuscript has been deemed suitable for publication in PLOS ONE. Congratulations! Your manuscript is now with our production department. 

Kind regards, 

on behalf of

Dr. Kuo-Cherh Huang 

Academic Editor

PLOS ONE